# Kinase Signaling in Colitis-Associated Colon Cancer and Inflammatory Bowel Disease

**DOI:** 10.3390/biom13111620

**Published:** 2023-11-06

**Authors:** Michelle Temby, Theresa L. Boye, Jacqueline Hoang, Ole H. Nielsen, John Gubatan

**Affiliations:** 1Division of Gastroenterology and Hepatology, Stanford University School of Medicine, Stanford, CA 94305-5101, USA; mtemby@stanford.edu (M.T.); jhoang10@stanford.edu (J.H.); 2Department of Gastroenterology, Medical Section, Herlev Hospital, University of Copenhagen, 2730 Herlev, Denmark; theresa.louise.boye@regionh.dk (T.L.B.); ole.haagen.nielsen@regionh.dk (O.H.N.)

**Keywords:** colon cancer, inflammation, kinase, inflammatory bowel disease, ulcerative colitis, Crohn’s disease, JAK inhibitor

## Abstract

Colorectal cancer is a known complication of chronic inflammation of the colon (“colitis-associated colon cancer”). Inflammatory bowel disease (IBD) is a chronic inflammatory disorder of the gastrointestinal tract. Patients with IBD are at increased risk of colon cancer compared to the general population. Kinase signaling pathways play critical roles in both the inflammation and regulating cellular processes such as proliferation and survival that contribute to cancer development. Here we review the interplay of kinase signaling pathways (mitogen-activated protein kinases, cyclin-dependent kinases, autophagy-activated kinases, JAK-STAT, and other kinases) and their effects on colitis-associated colon cancer. We also discuss the role of JAK-STAT signaling in the pathogenesis of IBD and the therapeutic landscape of JAK inhibitors for the treatment of IBD.

## 1. Introduction to Colitis-Associated Colon Cancer

Colorectal cancer (CRC) is the second leading cause of cancer death and the third most common malignancy according to the World Health Organization [1]. Patients with inflammatory bowel disease (IBD), i.e., ulcerative colitis (UC) and Crohn’s disease (CD), are at an increased risk of developing colitis-associated CRC (CAC) due to chronic colon inflammation [2]. UC increases the risk of CAC by 18–20%, while CD increases the risk by up to 8% after 30 years of active disease. IBD ranks as the third highest risk condition for CRC, with the top two spots being occupied by familial adenomatous polyposis (FAP) and hereditary nonpolyposis colorectal cancer syndrome (HNPCC) [3]. Furthermore, IBD patients with a family history of CRC are at an increased risk of developing CRC by at least two-fold when compared to IBD patients without a family history of CRC [2].

Prior mice models have provided evidence and mechanistic insights into the correlation between IBD and CAC [3,4]. A single injection of carcinogen azoxymethane (AOM) led to multiple colonic tumors but only in mice with chronic colitis. Multiple injections of carcinogen and a longer exposure time were required for tumors to form when inflammation was absent. It is important to note that CRC tumors did not arise in mouse models with previous inflammation; thus, it is unlikely that inflammation alone plays a decisive role in CRC initiation. However, in the case of CAC, “clinically detectable IBD always precedes (sometimes by decades) tumor initiation” [4]. Furthermore, the risk of CAC is directly correlated with the severity and longevity of active disease [4]. As mentioned, patients with IBD are at increased risk for developing CRC. Surprisingly, the overall incidence of IBD-associated CRC has been diminishing in recent decades in Western countries. UC incidence is reaching a plateau in Western countries; hence, colitis-associated colon cancer development may also be declining in Western countries [4]. However, in Asian countries, UC is still on the rise. The reasons for this rapid increase in the occurrence of IBD in Asian countries are not fully understood but may be related to increased industrialization and environmental changes in Asian countries (changes in sanitary conditions, exposure to infectious diseases, and the westernization of diet) [5]. Thus, it is expected that the occurrence of colitis-associated cancer worldwide will actually increase [6]. Previous studies have suggested that the risk of CRC in IBD increases with longer disease duration, the extent of colitis, a familial history of CRC, coexistent primary sclerosing cholangitis, and the degree of inflammation [7]. Notably, disease duration and colitis extent rank as the two most important risk factors in identifying IBD-associated CRC (Figure 1) [6]. However, the pathogenesis of CRC in IBD is poorly understood. It is widely assumed that IBD-associated CRC is a consequence of various genomic alterations and environmental factors (i.e., genetic instability, epigenetic alteration, immune response by mucosal inflammatory mediators, oxidative stress, intestinal microbiota). Constant colonic inflammation continues to be an important factor in CRC development [6]. Anti-inflammatory agents (i.e., 5-aminosalicylate compounds) and immune modulators work as potential chemopreventive agents for CRC. Regular colonoscopies are widely accepted as being effective in reducing the risk of colitis-associated CRC, although there is no clear evidence proving that surveillance colonoscopies prolong survival in patients with extensive colitis [6]. In this review, we first discuss the interplay of kinase signaling pathways and their effects on colitis-associated colon cancer. We then review JAK-STAT signaling in the pathogenesis of IBD and the therapeutic landscape of JAK inhibitors for the treatment of IBD.

## 2. Kinase Signaling in Colitis-Associated Colon Cancer

### 2.1. Mitogen-Activated Protein Kinases in CAC

Various kinase signaling pathways (i.e., mitogen-activated protein kinases (MAPKs), cyclin-dependent kinases, autophagy-activated kinases, JAK-STAT) have been proposed to contribute to colitis-associated colon cancer in various ways (Figure 2). Mitogen-activated protein kinases (p38α, p38β, p38γ, and p38δ) can promote CAC development through their role in regulating inflammation. Progress has been made by using p38α MAPK (also called p38) as a therapeutic target, but it has not gone past phase I/II clinical trials [8]. In determining the role of p38γ/p38δ in CRC development, a study conducted by Fajardo et al. treated p38γ/p38δ knockout mice with azoxymethane–dextran sodium sulfate (AOM-DSS) to recreate a CAC model in mice. Fajardo et al. [8] found that there is a positive correlation between p38γ expression and matrix metalloproteinase 9 (MMP9), tissue inhibitor of metalloproteinase 1 (TIMP1), and interleukin-6 (IL-6), which are overexpressed in CRC. They also found that IL-6 can act as a biomarker for colitis. In the same study, it was observed that p38γ is activated by inflammatory stimuli, which downstream leads to the production of cytokines and genes that are responsible for regulate cell survival. This observation suggests that p38γ has a role in the tumor microenvironment that is associated with tumor formation and progression [8]. In 30% of the tumors that were examined, the researchers noted higher levels of expression and activation of p38γ in CRC patients but not of p38δ expression levels [8]. p38γ protein levels were also considerably elevated in the plasma of patients with IBD, while p38δ and phosphorylated p38MAPK levels were lower in comparison to non-IBD patients. Because expression level and activation of p38γ is higher in CRC/IBD patients, it could be useful as a biomarker for colitis and early CRC diagnostic tests. Through the findings of the study, p38γ, when activated and at elevated levels, is reported to play a big role in CRC by increasing inflammation and promoting tumorigenesis [8].

### 2.2. Cyclin-Dependent Kinases in CAC

Cyclin-dependent kinases (CDKs), such as cyclin-dependent kinase 1 (CDK1), act as crucial operators of the cell cycle transition. Upregulation of CDK1 is associated with decreased survival time of patients with multiple cancer types, including colorectal, liver, and lung cancer [9]. Dysregulation of CDK1 causes rapid tumor growth and spontaneous proliferation of cancer cells, emphasizing that high gene expression levels of CDK1 are involved in the progression of multiple cancer types, including colorectal, liver, and lung cancer [9]. These high levels of CDK1 gene expression have been found to stimulate the proliferation of colorectal cancer cells [10]. CDK1 activity, such as p38γ, is highly elevated in CRC tissues compared to noncancerous tissues. High levels of CDK1 activity have been shown to be predictive of metastasis risk in CRC since CDK1 can promote CRC progression through phosphorylation of JAK1, activating the JAK/STAT3 signaling pathway. CDK1 inhibitors have been shown to stop the proliferation of colorectal cancer cells in in vitro and in vivo models [10]. A prior study [11] suggested that CDK1 signaling plays a mechanistic role in linking elevated intracellular iron levels with the development of colitis-associated colon cancer. The authors demonstrated that iron could bind to CDK1, which in turn promotes activation and sustained signaling of the oncogenic JAK1-STAT3 pathway in CRC and CAC [11]. 

### 2.3. Autophagy-Activated Kinases in CAC

Autophagy-activated kinases, such as Unc-51 and ULK1/2, regulate autophagy as a precisely controlled system consisting of five major stages: initiation, phagophore nucleation, autophagosome formation, lysosome fusion, and degradation [12]. The end goal of autophagy is to break down phagocytosed cellular components into amino acids with acidic hydrolases. The amino acids are then translocated to the cytoplasm to be reused [12]. Autophagy in epithelial cells is thought to have a protective role in IBD and colorectal tumorigenesis. However, multiple studies have also observed that epithelial autophagy may actually promote the progression of cancer and resistance to chemotherapy/radiotherapy. To promote cancer cell survival and proliferation, autophagy can activate kinase pathways such as JAK2/STAT3 and also suppress certain gene expressions such as p53 or p21 [13]. One study showed that inhibiting autophagy promoted p53 and endoplasmic reticulum stress-induced apoptosis, which resulted in a decreased tumor size [14]. As previously mentioned, chronic gut inflammation, such as in IBD, is a high-risk factor for colorectal cancer. This is important to note for autophagy-activated kinases because, while autophagy in intestinal epithelial cells inhibits colitis and colon carcinogenesis, it may also promote the development and progression of colon cancer. In established CAC, autophagy in colon cancer cells and the kinases that are used play a role in promoting tumorigenesis, which can lead to resistance of chemotherapy/radiotherapy. More so, autophagy in different cell types that exist in the cancer microenvironment, such as mesenchymal cells, dendritic cells, and immune cells, influences CAC development in different ways depending on the autophagy stage taking place, as well as the tumor microenvironment [15]. 

### 2.4. Janus Kinase Signaling in CAC

The last kinase signaling pathway we focus on is the Janus kinase–signal transducer and activator of transcription, better known as the JAK-STAT pathway. As with other kinases, inflammation activates JAK-STAT3 (STAT3) signaling pathways, which then induces the proliferation and remodeling of epithelial cells [16]. This ultimately promotes tumor development. In epithelial cells, STAT3 induces proteins that indirectly suppress apoptosis. For example, STAT3 induces chaperone protein Hsp70 and C-type lectin RegIIIβ, both of which are overexpressed in human colon cancer and IBD. In patients with IBD, STAT3 is activated by IL-6 as well as other cytokines and factors in the microenvironment; IL-11, IL-22, HGF, and EGF family members are active during the colitis stage, and may also be active during CAC/CRC tumorigenesis [16]. Inactivation of STAT3 in intestinal epithelium affects cell survival proliferation in colitis. However, in cancer models, inactivation of STAT3 actually leads to diminished CAC tumor growth [17]. Many inflammatory cytokines can activate STAT3 in epithelial cells due to coupled receptors of the JAK-STAT pathway. These receptors are expressed by immune cells as well as nonhematopoietic cells (such as stromal cells, normal epithelium, malignant epithelium). As part of the STAT3 pathway, IL-6 is a multifunctional cytokine produced by many cell types and is conducive to normal development of tissue regeneration, immune response, autoimmunity, and a multitude of cancers. Because of this, the effects IL-6 has on cancer cells of CAC and other forms of colorectal cancer are likely mediated by STAT3 [4,16].

### 2.5. Sphingosine 1-Phosphate (S1P) and Sphingosine Kinase 1 (SphK1)

Sphingosine 1-phosphate (S1P) functions to promote cell proliferation and cell survival, stimulate angiogenesis, and act both as an intracellular and extracellular inflammatory mediator [18,19]. S1P forms a complex with TNF-α receptor and TRAF2 to activate NF-κB signaling that controls DNA transcription, cytokine production, and cell survival [20]. S1P further regulates cell growth in different cells by increasing the expression of anti-apoptotic Bcl-2 and Mcl-1 and downregulating the pro-apoptotic proteins BAD and BAX [21,22]. Sphingosine kinase 1 (SphK1) and sphingosine kinase 2 (SphK2) phosphorylate sphingosine to synthesize S1P. Persistent activation of STAT3 (signal transducer and activator of transcription 3) is associated with S1P receptor signaling. Interestingly, STAT3-mediated signaling pathways promote the activation of SphK1 and SphK2 and lead to the production of S1P through a positive autocrine loop [19]. Some studies have further examined the role of STAT3 as an important link between inflammation and cancer [19]. Their results indicate that constant activation of STAT3 in epithelial cells in inflammation-associated CRC is associated with S1P receptor signaling, which indicates that the SphK1/S1P pathway has an impact on different pathways associated with CAC [19]. A study conducted by Kawamori et al. [18] investigated the role of SphK1/S1P in regulating cyclooxygenase-2 (COX-2), which is an established pathogenic factor in colon carcinogenesis. The results of the study suggest that the SphK1/S1P pathway is involved with COX2 expression of colon carcinogenesis using in vivo models [18]. In another study conducted by Kawamori et al. [23], the role of the SphK1/S1P pathway was investigated in AOM/DSS-induced colon carcinogenesis. The results showed that SphK1 is upregulated in human colon carcinogenesis and that SphK1 knockout mice were significantly protected in the early and late stages of colon carcinogenesis [23]. Additionally, they found that SphK1 is overexpressed in human colon tumors, including adenomas and adenocarcinomas, and that SphK1 expression is higher in primary colon cancers with metastases than in those without [24]. They also found that SphK1 deficiency significantly inhibited AOM-induced ACF formation, which is a preneoplastic lesion of colon cancer in a model of progressive malignant development, and that SphK1 deficiency significantly inhibited AOM/DSS-induced COX-2 expression [24]. This suggests that the SphK1/S1P pathway may mediate its expression as a downstream target in inflammation-mediated CRC [24]. 

As a target for therapy, the oral drug, ozanimod, is the first S1P receptor modulator to be approved by the FDA for the treatment of moderate to severe UC, but long-term efficacy and safety data have yet to be accumulated [25]. Ozanimod targets S1P receptor subtypes 1, 4, and 5, and it works to downregulate S1PRs on lymphocytes and inhibit their migration from lymphoid organs to sites of inflammation [25]. In a recent phase III clinical trial, ozanimod was effective in its treatment of patients with moderate to severe UC [26].

### 2.6. Death-Associated Protein Kinase (DAPK)

Death-associated protein kinase (DAPK) functions as a tumor suppressor by regulating apoptosis, autophagy, membrane blebbing, and stress fiber formation. The expression of DAPK is downregulated in many types of cancer by methylation of the DAPK promoter. DAPK both promotes and inhibits inflammation depending on signals in its microenvironment [27]. DAPK is inactivated by methylation, and this has been related to different types of cancers including inflammation-associated tumors and their premalignant lesions such as chronic gastritis—gastric cancer. In this study [27], DAPK hypermethylation was found to be associated with the inflammation of mucosa in UC patients, and higher methylation was directly correlated to higher inflammation [28]. When looking at the methylation of DAPK in UC and UC-associated carcinoma, a study conducted by Kuester et al. analyzed promoter hypermethylation and protein expression of DAPK and found that promoter methylation correlated significantly with decreased DAPK protein expression. DAPK protein expression also increased with the severity of inflammation in UC. For methylated cases, it was observed that there was more inflammatory activity compared to unmethylated samples [29]. However, compared to sporadic colorectal carcinoma, this study found that UC-associated colorectal cancer had a significantly lower methylation frequency [18]. Further studies should be initiated to investigate the potential role of DAPK promoter hypermethylation in IBD. 

### 2.7. Myosin Light Chain Kinase (MLCK)

Tight junction dysregulation and epithelial damage contribute to barrier loss in patients with IBD [30]. Myosin light chain kinase (MLCK) plays a role in intestinal epithelial barrier function through its influence on tight junctions. MLCK is activated by tumor necrosis factor receptor 2 (TNFR2), which is a major inflammatory cytokine receptor that is also involved in the pathogenesis of IBD. Its respective ligand tumor necrosis factor (TNF) also plays a role in contributing to CAC development through its activation of NFkB in myeloid and intestinal cells [31]. TNFR2 activates MLCK-dependent tight junction dysregulation and causes intestinal barrier loss via apoptosis, and thus induces colitis [30]. A study conducted by Su et al. [30] found that in the early stages of colitis, MLCK-dependent tight junction permeability increases, but in advanced stages of colitis, it progresses by apoptosis and mucosal damage that result in tight-junction-independent barrier loss and barrier loss that is independent of MLCK, which suggest that there are two temporally distinct mechanisms [30]. To further look at the downstream effects of MLCK on colitis-associated CRC, studies have found that suppression of MLCK as well as blockage of TNFR2 signaling restore tight junctions, decrease pro-tumorigenic cytokines, and reduce CAC development. A study conducted by Suzuki et al. suggests MLCK is a potential target for CAC prevention [32]. Here, increased levels of pro-tumorigenic cytokines (IL1β, IL-6, and MIP2) in animal models of colitis and animal models of CAC are associated with upregulated MLCK expression [32]. These results indicate that an increase in cytokine-producing infiltrating cells in tissues may be required for CAC progression and that these cytokines might be associated with disrupted epithelial tight junctions [32]. 

### 2.8. Epidermal Growth Factor Receptor 2 (EGFR2) and HER2 (Receptor Tyrosine-Protein Kinase erbB-2)

Within the EGF receptor family of proteins (ErbB), we next discuss epidermal growth factor receptor (EGFR) and receptor tyrosine-protein kinase ErbB-2 (HER2) as they relate to IBD and CAC. EGFR is a receptor tyrosine kinase that requires ligand binding to activate its tyrosine kinase domain. It is involved in activating signaling pathways responsible for cell proliferation, angiogenesis migration, continued existence, and adhesion [33]. EGFR is known to activate a cascade of multiple signaling pathways to help facilitate tumor growth processes, so it would be an interesting target to investigate in the context of colorectal cancer. Interestingly, the basal level expression of EGFR2 is higher in colorectal cancer tissue than in the surrounding mucosa [34]. The exact mechanism of how EGFR2 contributes to the pathogenesis of colitis and CAC is unclear. HER2 is commonly known for its role in signaling pathways that control cell proliferation, survival, and apoptosis in breast cancer. In the context of colon cancer, HER2 activation can lead to the loss of epithelial cell tight junctions, which is also observed in the pathogenesis of IBD. HER2 may also play a role in promoting and/or maintaining the loss of polarity of epithelial cells, which suggests a role in carcinogenesis [35]. About 7% of CRC patients have HER2 somatic mutations or HER2 gene amplifications. HER2-activating mutations cause EGFR resistance in colorectal cell lines [36]. The significance of HER2 in IBD requires further exploration.

### 2.9. Phosphoinositide 3-Kinase (PI3K)

The phosphoinositide 3-kinase (PI3K)/Akt pathway is one of the intracellular pathways activated downstream from IL-6 activation. IL-6 plays a crucial role in IBD, and it has been shown that patients with IBD have elevated levels of IL-6, which is positively correlated with severity of inflammation [32]. The PI3K/Akt pathway is also activated in tumorigenesis. PI3K is a lipid kinase that generates PIP3 and is reserved by PTEN. Akt is translocated to the plasma membrane via secondary messenger PIP3 to be phosphorylated and activated by PDK1. Once activated, Akt is involved in cellular functions such as cell survival and proliferation. PI3K/Akt expression has positive effects on cell survival. A study found that the PI3K/Akt pathway is involved with inflammation-associated tumorigenesis and works in parallel with the Ras/Erk pathway to generate cell survival and anti-apoptotic cell signals [37]. This study used animal models for UC, CRC, and CAC using DSS and 1,2 dimethyl hydrazine (carcinogenic agent) in BALB/c mice. The study looked at levels of several components of the PI3K/Akt/PTEN pathway (PI3k, Akt, mTOR, PDK1, and beta catenin) in CAC and found that they were elevated in the DSS, DMH, and DSS + DMH groups, which suggests that this pathway is involved in inflammation-mediated tumorigenesis [37].

## 3. JAK-STAT Signaling

The JAK-STAT signaling pathway orchestrates cell signaling through intracytoplasmic protein-tyrosine kinases that bind the cytoplasmic region of transmembrane cytokine receptor subunits upon cytokine activation. The JAK-STAT signaling pathway consists of type I and type II cytokine receptors, four JAK family members (JAK1, JAK2, JAK3, and tyrosine kinase 2 (TYK2)), along with seven STAT family members (STAT1–4, STAT5a/b, and STAT6). Type II receptors (interferons (IFNs) (type 1, -2 or -3) and IL-10 receptor families) are distinguished from type I receptors (gp130, IL-12, and common γ- and β-chain receptor families) by the lack of a WSXWS motif—a molecular switch involved in receptor activation [38]. The STAT-JAK signaling pathway regulates numerous important cellular functions, including adaptive and innate immunity, hematopoiesis, cell division, differentiation, and migration [39] (Figure 3).

## 4. JAK Inhibitors for IBD

### 4.1. The JAK System

A paradigm shift had occurred in the management strategy of IBD by the end of the last century with the introduction of the first biologic, infliximab, a TNF inhibitor [42]. Since then, this biologic agent has been followed by other TNF inhibitors: adalimumab, certolizumab pegol, and golimumab, as well as the anti-integrin, vedolizumab, and more recently, the anti-cytokine to the common p40 subunit of IL-12/-23 [43] or the p19 subunit of IL-23 [44,45]. However, an unmet need still exists for therapeutics of IBD since patients may lose the effect to biologics (i.e., as primary nonresponders or secondary nonresponders) [42]. Therefore, pharmacological research has continued within the drug class of small molecules comprising Janus kinase (JAK) inhibitors [46] and sphingosine-1-phosphate agonists [47]. As compared with biologics, small molecules have advantages, such as being orally administered (i.e., the use of specialized staff for dispensing the drug is avoided), being without risk of immunogenicity, having predictable pharmacogenetics, having a rapid onset of action, being more economic to produce than parenteral drugs, being easy to store at an ambient temperature, and having a short half-life of importance in case of, e.g., surgery or infections [46]. 

The JAK system was first described in the 1990s, but due to novel knowledge about the pathophysiology of IBD revealing the influence of cytokines in the inflammatory process [48], a major development occurred in 2018 with the introduction of the pan-JAK inhibitor, tofacitinib, the first drug in its class of small molecules used for the management of IBD [46]. Tofacitinib, however, targeting all four isoforms of cytokine-associated JAKs (i.e., JAK1, JAK2, JAK3, and TYK2), some of which are believed to be beneficial for the physiology of humans, may influence the risk of adverse events related to this drug [46].

### 4.2. Pan-JAK Inhibitor for IBD

Although the first-generation pan-JAK inhibitor, tofacitinib, has been demonstrated to be a highly effective drug for both first- and second-line therapy of ulcerative colitis [49,50], but not for Crohn’s disease [51], concerns about its safety, including the risk of infection, venous thromboembolism, major adverse cardiovascular events, and malignancy, have, however, dampened enthusiasm for its widespread use. 

The safety concerns for tofacitinib were detailed in a post hoc analyses of randomized controlled trials (RCTs) with 8524 patients exposed to JAK inhibitors [52]. The safety data showed an overall relative risk (RR) of nonmelanoma skin cancer of 1.05 (95% CI 0.47–2.35), and when excluding studies with active comparator arms, the RR was 1.22 (95% CI 0.50–2.95) [52]. Moreover, this study found in a pooled analysis of 21 RCTs, including 9916 patients exposed to JAK inhibitors, an overall RR for malignancy of 1.39 (95% CI 0.68–2.85), and when considering solely placebo-controlled RCTs, an RR of 1.50 (95% CI 0.68–3.32) was found [52]. Moreover, another large postmarketing safety study of tofacitinib versus TNF inhibitors for various diseases (mainly administered for rheumatoid arthritis but also other diseases such as ulcerative colitis), the ORAL Surveillance phase IV noninferiority clinical trial of 4361 patients also evaluated the risk of malignancies (excluding NMSC) in patients using tofacitinib 5 mg b.i.d. (n = 1455 patients) or 10 mg b.i.d. (n = 1456 patients), or a TNF antagonist (n = 1451 patients) [53]. Overall, the hazard ratio (HR) of malignancies for tofacitinib 5 mg b.i.d. or 10 mg b.i.d. versus a TNF antagonist were 1.47 (95% CI 1.00–2.18) and 1.48 (95% CI 1.00–2.19), respectively, not achieving the noninferiority criteria for either the tofacitinib dose of 5 mg b.i.d. or that of 10 mg b.i.d. [53]. 

Adverse drug reactions are thought to be more challenging when inhibiting multiple JAKs simultaneously, and thus, selective JAKs are presumably safer than pan-JAK inhibitors. Nevertheless, the harm–benefit balance of selectively inhibiting JAK1, JAK2 JAK3, and/or TYK2 has still not been fully elucidated. For example, completely inhibiting JAK2 may possibly lead to anemia and other unacceptable cytopenic effects, including thrombocytopenia [54], whereas inhibiting JAK3, which is generally restricted to the immune system, can increase the risks of bacterial and viral infections [55]. Thus, selective inhibition of JAK2 or JAK3 may cause a number of side effects as these two JAKs are crucial for homeostasis in humans. Additionally, targeting JAK1 has been shown to cause dose-dependent and transient lipid elevation in the circulation [56], but it is unknown if lipid changes translate into cardiovascular disease. On the other hand, TYK2 mainly inhibits proinflammatory cytokines, and thus, selectively targeting TYK2 may be of greater potential therapeutic interest in immune-mediated diseases such as IBD, Therefore, it is speculated that selective avoidance of excessive inhibition of particularly JAK2 and JAK3 may result in a more optimal safety profile as compared with inhibiting all four JAKs (JAK1-3 and TYK2) using a pan-JAK inhibitor.

Based on the safety data mentioned above, the US FDA has issued a class-wide boxed warning based on not only malignancies, but also elevated rates of serious infections and thromboembolism observed with tofacitinib, 5 mg or 10 mg twice per day [53]. It should, however, be noticed that there is no explanation for the mechanism behind the risk of thrombosis so far. This may be due to a combination of multiple factors, including JAK inhibition and cytokines, as well as the consequences of inflammation from immune-mediated disease [57].

### 4.3. Selective JAK Inhibitors for IBD

Due to concerns about the safety profiles based on the experience with the pan-JAK inhibitor, tofacitinib, focus has been put on the development of more selective JAK inhibitors, including upadacitinib and filgotinib [58,59], which preferentially target JAK-1 (Figure 4). Both drugs are already marketed for IBD with an expected more beneficial safety profile. However, due to male reproductive issues of filgotinib in preclinical animal studies (impaired spermatogenesis and histopathologic effects on male reproductive organs (testis and epididymis)) with dosages needed for IBD, this drug has been approved by EMA only, and is not expected to be introduced in the US for IBD anytime soon [60,61]. Moreover, other JAK inhibitors preferentially targeting TYK2 are in late-stage clinical trials and are expected to be marketed within the next years for this indication [62]. The available clinical studies for both upadacitinib [63,64] and filgotinib [65,66] have not reported an elevated risk for malignancies compared to nonexposed patients, but the observation time has been limited as these studies were not powered for safety analysis [56]. Moreover, preliminary data reported on TYK2 inhibitors in clinical studies for IBD have been convincing, although long-term data are missing as well [62]. 

Other JAK inhibitors have also been examined in patients with IBD. Izencitinib (TD-1473) is an oral gut-selective pan-JAK inhibitor. In a previous phase 1b randomized study with moderately to severely active ulcerative colitis (ClinicalTrials.gov NCT02657122, NCT02818686), patients received once-daily oral TD-1473 20, 80, or 270 mg, or a placebo for 28 days. The study demonstrated that gut-selective pan-JAK inhibition with izencitinib administration resulted in high intestinal vs plasma drug exposure, local target engagement, and trends toward reduced UC disease activity [67]. However, izencitinib failed to meet its primary endpoint in a subsequent phase IIb dose-finding induction clinical trial to treat ulcerative colitis [68]. Brepocitinib (PF-06700841) is a selective JAK1 and TYK2 inhibitor with IC50 values of 17 and 23 nM, respectively [69]. A phase IIb 32-week induction/maintenance clinical trial enrolled patients with active UC (total Mayo Score of ≥6), who were randomly chosen to receive oral brepocitinib (10, 30, or 60 mg q.d.) or a placebo. In this study, patients also received one of three doses of ritlecitinib (PF-06651600) (a JAK3/TEC inhibitor; 20, 70, or 200 mg q.d.) or a placebo for 8 weeks. At week 8, the proportions of participants that were in remission (total Mayo Score of ≤2, no individual subscore of >1, and rectal bleeding subscore of 0) were 8.3%, 23.4%, and 23.4% for brepocitinib 10, 30, and 60 mg q.d., respectively, and 9.8%, 28.6%, and 34.0% for ritlecitinib 20, 70, and 200 mg q.d., respectively, versus 0% for the placebo group [70]. Brepocitinib is currently being tested with ritlecitinib on patients with CD (ClinicalTrials.gov identifier NCT03395184). 

Opportunistic infections have been reported with JAK inhibitors at a rate of 0.1–0.3 per 100 patient years [71]. Observations include multidermatomal herpes zoster, esophageal candidiasis, pneumocystitis, CMV, and cryptococcal infection. The rate of these infections seems similar to those reported for patients on biologics [72], but there is a greater risk of developing herpes zoster associated with older age and higher prednisolone dose [73]. This risk appears to be increased for both nonselective and selective JAK inhibitors, which suggests a need to establish vaccination strategies for prevention across all drugs of this class [52,64]. The current recommendation suggests patients be vaccinated for herpes zoster 2–4 weeks before initiating a JAK inhibitor [72].

**Figure 4 biomolecules-13-01620-f004:**
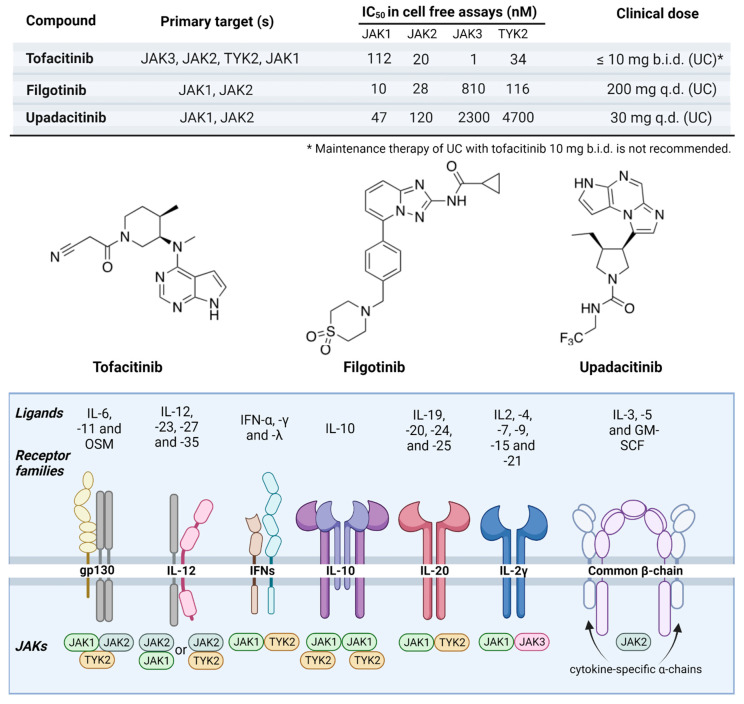
JAK inhibitors used in IBD. The JAK inhibitors currently used in the treatment of IBD include tofacitinib, filgotinib, and upadacitinib. Here, we show their specificity to effector JAK proteins, clinical dose, and chemical structures, and an overview of the various affected receptor families. Most cytokine receptor families are characterized by forming heterodimers of two different polypeptide chains, each of which binds a different JAKs or TYK2. For example, OSM utilize the gp130/OSM receptor heterodimer complex for signaling. Nevertheless, some receptors are tetrameric consisting of four structural subunits. For example, IL-6, and IL-11 receptors initially homodimerize before associating with the gp130 homodimer complex, and the IL-10 receptor is comprised of both 2α and 2β subunits. Oher cytokine receptor families are hexameric consisting of six structural subunits. For example, the GM-CSF receptor is comprised of two α chains (GMRα), two intercalated β chains (β common chains), and two GM-CSF molecules. The various combinations of receptor chains utilize different members of the JAK and TYK2 kinases that downstream functions as proinflammatory transcription factors. Selectivity data for jakinibs were obtained from [74].

## 5. Future Perspectives

In general, the incidence rates for adverse events of JAK inhibitors are considered to be dose-dependent resulting in hesitations dealing with doses of tofacitinib exceeding 5 mg b.i.d., filgotinib (200 mg q.d.), and upadacitinib (15 mg q.d.). Consequently, it is recommended to limit the induction dosage of tofacitinib 10 mg b.i.d. in UC for up to 16 weeks as a maximum—preferably 8 weeks or less for those patients for whom no suitable alternative treatment exists. Due to the complexity of the JAK-mediated intracellular signaling pathways, it is important to longitudinally monitor patients included in registry studies to determine if there are any long-term consequences. Moreover, JAK inhibitors are not yet recommended during pregnancy due to sparse evidence and the observation of teratogenic effects in preclinical animal studies [75]. Given that the JAK-STAT signaling pathway also plays a role in colitis-associated colon cancer, longitudinal studies will be needed to determine whether JAK inhibitors and targeting their downstream signaling pathways (e.g., STAT3) [76] may also reduce the risk of colon cancer among patients with IBD in addition to treating luminal inflammation. 

## Figures and Tables

**Figure 1 biomolecules-13-01620-f001:**
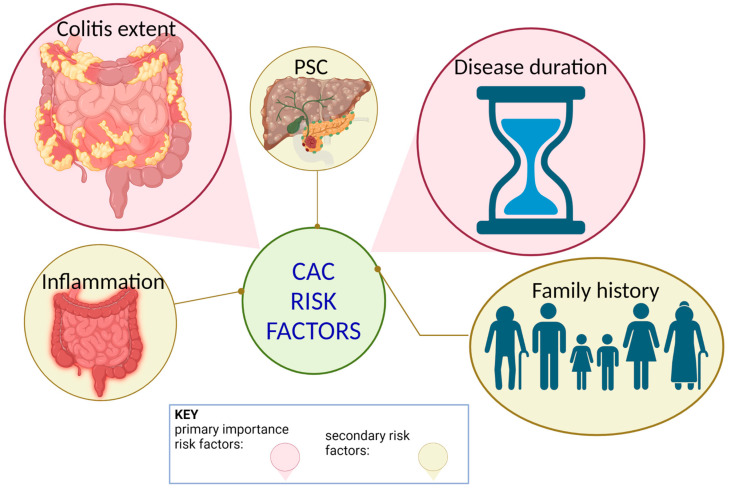
Colitis-associated colitis risk factors. Previous studies have suggested that the risk of CAC increases with longer duration of disease, the extent of colitis, a familial history of CRC, coexistent primary sclerosing cholangitis (PSC), and the degree of inflammation. Disease duration and colitis extent rank as the two most important risk factors in identifying IBD-associated CRC.

**Figure 2 biomolecules-13-01620-f002:**
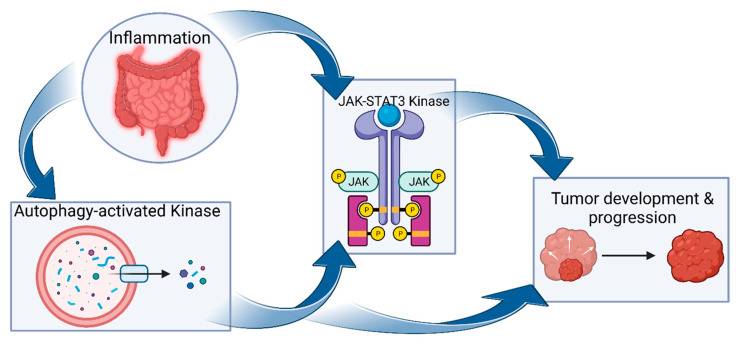
Autophagy-activated kinases and JAK-STAT3 interplay in CAC. Autophagy may promote cancer cell proliferation and survival by activating other kinase pathways, such as those of JAK2/STAT3 pathways. As with other kinases, inflammation—such as that in IBD—activates JAK-STAT3 (STAT3) signaling pathways. These kinase pathways induce epithelial cell proliferation and remodeling, ultimately promoting tumor development.

**Figure 3 biomolecules-13-01620-f003:**
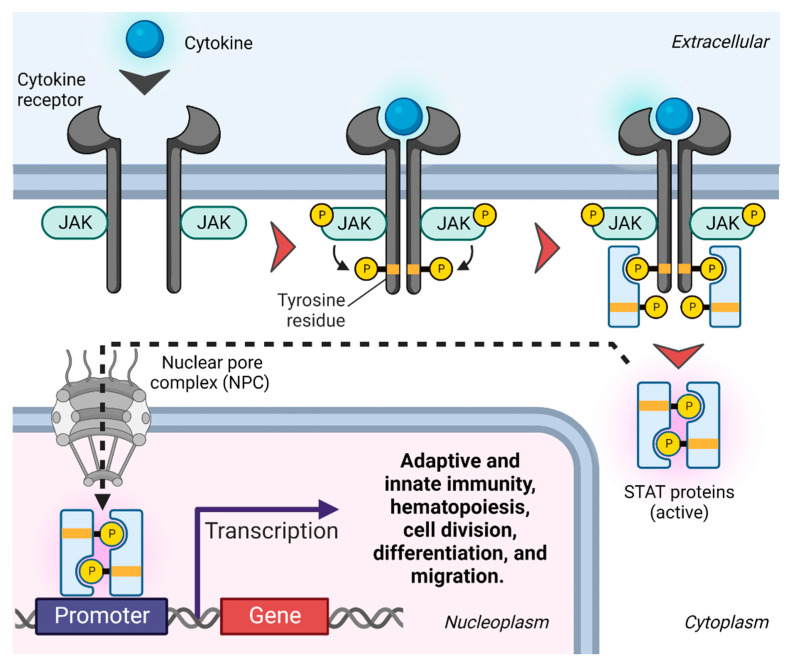
JAK-STAT signaling. Upon ligand-mediated receptor dimerization, JAKs are phosphorylated and activated, which in turn induces tyrosine phosphorylation of the cytosolic receptor tails. Receptor phosphorylation forms docking sites for receptor-specific STAT proteins, which are subsequently activated by phosphorylation. Activated STAT proteins form homo- or heterodimers that pass through the nuclear membrane through nuclear pore complexes (NPCs) [40] and successively bind to specific enhancer sequences in target genes, impacting their transcription [41].

## Data Availability

Not applicable.

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
