# Peer review of "Kinase Signaling in Colitis-Associated Colon Cancer and Inflammatory Bowel Disease"

_biomolecules, 2023, doi:10.3390/biom13111620_

Round 1

Reviewer 1 Report

Comments and Suggestions for Authors

I have the following comments and suggestions:

1.       Lines42-46, “Surprisingly, the overall incidence of IBD-associated CRC has been diminishing in recent decades in western countries. UC incidence is reaching a plateau in Western countries, hence colitis-associated colon cancer development may also be declining in Western countries. However, in Asian countries, UC is still on the rise.” Can the authors speculate what factors may cause this different trend in Western vs Asian countries?

2.       Section 2.2, suggest to add in this reference (“Iron uptake via DMT1 integrates cell cycle with JAK-STAT3 signaling to promote colorectal tumorigenesis.” PMID: 27546461) to discuss the integration mechanism for CDK1 and JAK/STAT3 pathways in CRC.

3.       Line 247, “The JAK this system…”, “this” should be removed.

4.       Despite that agents targeting STAT3 are still in the early phases of development, since STAT3 is proved as a key regulator of tumor-promoting inflammation as well as CRC resistance to chemoradiotherapy, it may be pertinent to add in a paragraph for STAT3 inhibitor after section 4.3, referring to a recent review (“Targeting STAT3 signaling pathway in colorectal cancer.” PMID: 34440220).

Author Response

REVIEWER 1

  1. Lines42-46, “Surprisingly, the overall incidence of IBD-associated CRC has been diminishing in recent decades in western countries. UC incidence is reaching a plateau in Western countries, hence colitis-associated colon cancer development may also be declining in Western countries. However, in Asian countries, UC is still on the rise.” Can the authors speculate what factors may cause this different trend in Western vs Asian countries?

Response: Thank you for allowing us to clarify. We have added the following to the manuscript: “The reasons for this rapid increase in the occurrence of IBD  in Asian countries are not fully understood but may be related to increased industrialization and environmental changes in Asian countries (changes in sanitary conditions, exposure to infectious dis-eases and the westernization of diet.” (Mak WY, Zhao M, Ng SC, Burisch J. The epidemiology of inflammatory bowel disease: East meets west. J Gastroenterol Hepatol. 2020 Mar;35(3):380-389. doi: 10.1111/jgh.14872. Epub 2019 Nov 24. PMID: 31596960.)

  1. Section 2.2, suggest to add in this reference (“Iron uptake via DMT1 integrates cell cycle with JAK-STAT3 signaling to promote colorectal tumorigenesis.” PMID: 27546461) to discuss the integration mechanism for CDK1 and JAK/STAT3 pathways in CRC.

Response: Thank you for this excellent recommendation. We have added the cited study above, “ A prior study [11] suggested that CDK1 signaling plays a mechanistic role in linking elevated intracellular iron levels with the development of colitis-associated colon cancer. The authors demonstrated that iron could bind to CDK1 which in turn promotes activation and sustained signaling of the oncogenic JAK1-STAT3 pathway in CRC and CAC [11].”

  1. Line 247, “The JAK this system…”, “this” should be removed.

Response: This has been removed.

  1. Despite that agents targeting STAT3 are still in the early phases of development, since STAT3 is proved as a key regulator of tumor-promoting inflammation as well as CRC resistance to chemoradiotherapy, it may be pertinent to add in a paragraph for STAT3 inhibitor after section 4.3, referring to a recent review (“Targeting STAT3 signaling pathway in colorectal cancer.” PMID: 34440220).

Response: Focusing on STAT signaling is beyond the scope of this review and remains a future direction for targeting this pathway for colitis-associated colon cancer. Nonetheless, we cite the above referenced paper as a future direction: “Given that the JAK-STAT signaling pathway also plays a role in colitis-associated colon cancer, longitudinal studies will be needed to determine whether JAK inhibitors and targeting their downstream signaling pathways (e.g. STAT3) [62] may also reduce the risk of colon cancer among patients with IBD in addition to treating luminal inflammation.”

Reviewer 2 Report

Comments and Suggestions for Authors

The present manuscript describes as a review the role of the kinase signal
pathway in the development of colitis-associated colon cancer. Some inflammatory
cytokines, such as IL-9 or IL-23 as well as interferon-g, depend on Janus kinases
(they were originally named "Just Another Kinase") and result in the phosphorylation
of signal transducing and activating transcription factors such as STAT molecules.
After the STAT proteins are phosphorylated, they enter the cell nucleus and regulate
gene transcription, leading to CAC.
 -The authors first give a rough overview and show in Figure 1 a fairly general
illustration that is rather trivial. Here the causes should be presented more
scientifically than, for example, an hourglass or various pictograms…
 -In Chapter 2, the authors try to give an overview of the kinase signaling pathways,
but limit themselves to only a limited view. For example, SphK1 or DAPK or MLCK are
missing as kinases, which also pose a not inconsiderable risk in the development of
CAC. Another chapter should be added here. Sphingosine 1-phosphate (S1P) in
particular is a bioactive lipid mediator produced by sphingosine kinase 1 (SphK1)
and is known to play an important role in inflammation and cancer progression.
In addition, SphK1 and S1P function as upstream mediators of the pro-inflammatory
cytokine interleukin 6 (IL-6) and the signal transducer and activator of
transcription-3 (STAT3). The cytoskeleton-associated serine/threonine kinase
death-associated protein kinase (DAPK) has been previously described as a
cancer gene chameleon with functional antagonistic duality in a cell-type and
context-specific manner. Finally, it has been reported that myosin light chain
kinase (MLCK) is responsible for epithelial permeability associated with TNF
signaling. Inhibition of MLCK coupled with blockade of TNFR2 signaling resulted
in decreased CAC development.
 -Chapter 4 describes the possible inhibitors for JAK, but is again limited to the
three known candidates. However, an overview should offer the reader the most
up-to-date status possible, and in this case the two future promising candidates
should also be included. The active ingredient TD-1473 in particular belongs to
the group of pan-JAK inhibitors that is currently being investigated in a phase Ib study for patients with moderate to severe UC. Brepocitinib (PF-06700841), an inhibitor of JAK 1 and TYK2, and the JAK 3 inhibitor PF-06651600 are also being investigated in a phase IIb study in moderate to severe UC. These should still be listed.
For this reason I propose a major revision.  

Author Response

REVIEWER 2

The present manuscript describes as a review the role of the kinase signal pathway in the development of colitis-associated colon cancer. Some inflammatory cytokines, such as IL-9 or IL-23 as well as interferon-g, depend on Janus kinases (they were originally named "Just Another Kinase") and result in the phosphorylation of signal transducing and activating transcription factors such as STAT molecules. After the STAT proteins are phosphorylated, they enter the cell nucleus and regulate gene transcription, leading to CAC.  -The authors first give a rough overview and show in Figure 1 a fairly general illustration that is rather trivial. Here the causes should be presented more scientifically than, for example, an hourglass or various pictograms.In Chapter 2, the authors try to give an overview of the kinase signaling pathways, but limit themselves to only a limited view. For example, SphK1 or DAPK or MLCK are missing as kinases, which also pose a not inconsiderable risk in the development of CAC. Another chapter should be added here. Sphingosine 1-phosphate (S1P) in particular is a bioactive lipid mediator produced by sphingosine kinase 1 (SphK1) and is known to play an important role in inflammation and cancer progression. In addition, SphK1 and S1P function as upstream mediators of the pro-inflammatory cytokine interleukin 6 (IL-6) and the signal transducer and activator of transcription-3 (STAT3). The cytoskeleton-associated serine/threonine kinase death-associated protein kinase (DAPK) has been previously described as a cancer gene chameleon with functional antagonistic duality in a cell-type and context-specific manner. Finally, it has been reported that myosin light chain kinase (MLCK) is responsible for epithelial permeability associated with TNF signaling. Inhibition of MLCK coupled with blockade of TNFR2 signaling resulted in decreased CAC development. 

Response: Thank you for the valuable feedback. We have revised the manuscript to include additional kinases (Sphingosine 1-phosphate (S1P) and sphingosine kinase 1, Death-associated protein kinase, Myosin light chain kinase, Epidermal Growth Factor Receptor 2 (EGFR2), HER2, Phosphoinositide 3-kinase) under Section 2.

Chapter 4 describes the possible inhibitors for JAK, but is again limited to the three known candidates. However, an overview should offer the reader the most up-to-date status possible, and in this case the two future promising candidates should also be included. The active ingredient TD-1473 in particular belongs to the group of pan-JAK inhibitors that is currently being investigated in a phase Ib study for patients with moderate to severe UC. Brepocitinib (PF-06700841), an inhibitor of JAK 1 and TYK2, and the JAK 3 inhibitor PF-06651600 are also being investigated in a phase IIb study in moderate to severe UC. These should still be listed. For this reason I propose a major revision. 

Response: Thank you for the feedback. We initially only included FDA approved or late clinical trial stage JAK inhibitors in the review. However,  we have add the additional JAK inhibitors recommended by the reviewer: “Other JAK inhibitors have also been examined in patients with IBD. Izencitinib (TD-1473) is an oral gut-selective pan-JAK inhibitor. In a previous, Phase 1b randomized study with moderately to severely active ulcerative colitis (Clinicaltrials.gov NCT02657122, NCT02818686), patients received once-daily oral TD-1473 20, 80 or 270 mg, or placebo for 28 days. The study demonstrated that gut-selective pan-JAK inhibition with TD-1473 ad-ministration resulted in high intestinal vs plasma drug exposure, local target engagement, and trends toward reduced UC disease activity [71]. A follow-up, Izencitinib (TD-1473) failed to meet its primary endpoint in Phase IIb dose-finding induction clinical trial to treat ulcerative colitis [72]. Brepocitinib (PF-06700841) is a selective JAK1 and TYK2 inhib-itor with IC50 values of 17 and 23nM, respectively [73].  A Phase IIb 32-week induc-tion/maintenance clinical trial enrolled patients with active UC (total Mayo Score of ≥6), who were then randomized to receive oral Brepocitinib (10, 30, or 60 mg qd) or placebo. In the same study, patients also received one of three doses of Ritlecitinib (PF-06651600) (a JAK3/TEC inhibitor; 20, 70, or 200 mg qd) or placebo for 8 weeks. At week 8, the propor-tions of participants achieving remission(total Mayo Score of ≤2, no individual subscore of >1, and rectal bleeding subscore of 0) were 8.3%, 23.4%, and 23.4% for Brepocitinib 10, 30,and 60 mg qd, respectively, and 9.8%, 28.6%, and 34.0% for Ritlecitinib 20, 70, and 200 mg qd, respectively, versus 0% for placebo[74]. Brepocitinib is currently being tested with Ritlecitinib on patients with CD (ClinicalTrials.gov identifier NCT03395184).

Reviewer 3 Report

Comments and Suggestions for Authors

This manuscript is regarding to review the kinase signaling in colitis related colon cancer. The authors aimed to described several kinase signaling in colitis associated colon cancer. The structure and description is clear and sound. There are some points may be mentioned.

1.     why the authors choose these kinases. The other kinase signaling, such as EGFR, PI3K, or HER2…, has role in colitis related colon cancer.

2.     Could the authors provide a table to summary the receptors, kinase, and target (genes, cytokines, or cytoskeletal changes….).

3.     The process of colitis to colon cancer, as adenoma to colon cancer, and the role of kinase may be mentioned more clearly. May a figure is considered.

Author Response

REVIEWER 3

This manuscript is regarding to review the kinase signaling in colitis related colon cancer. The authors aimed to described several kinase signaling in colitis associated colon cancer. The structure and description is clear and sound. There are some points may be mentioned.

  1. why the authors choose these kinases. The other kinase signaling, such as EGFR, PI3K, or HER2…, has role in colitis related colon cancer.

Response: Thank you for the feedback. This was meant to be a focused review of the major kinase signaling pathways in CAC. However, given the recommendations from reviewers, we have included additional kinases (Sphingosine 1-phosphate (S1P) and sphingosine kinase 1, Death-associated protein kinase, Myosin light chain kinase, Epidermal Growth Factor Receptor 2 (EGFR2), HER2, Phosphoinositide 3-kinase) under Section 2.

  1. Could the authors provide a table to summary the receptors, kinase, and target (genes, cytokines, or cytoskeletal changes….).

Response: This review was meant to be a descriptive and narrative overview of the various kinase signaling pathways implicated in CAC and not a cataloging (via table) of all the genes, receptors, and target cytokines in all the kinase pathways mentioned. The requested table is excessive and not including the table does not detract from the central message of this invited review.  

  1. The process of colitis to colon cancer, as adenoma to colon cancer, and the role of kinase may be mentioned more clearly. May a figure is considered.

Response: The process and mechanisms of colitis progressing to colon cancer was described in great detail in the context of kinase signaling pathways throughout this review.  Our figure describes a broad schematic of inflammation-associated pathways that could lead to CAC. The progression of sporadic adenoma to colon cancer is beyond the scope of this review.

Round 2

Reviewer 1 Report

Comments and Suggestions for Authors

The authors addressed most of my comments.

Author Response

Thank you for the feedback.

Reviewer 2 Report

Comments and Suggestions for Authors The authors have now revised all of my critical points and turned the manuscript into an easy to read and informative article. thank you very much

Author Response

Thank you for the feedback.